# Influence of High-Intensity Interval Training on IGF-1 Response, Brain Executive Function, Physical Fitness and Quality of Life in Sedentary Young University Women—Protocol for a Randomized Controlled Trial

**DOI:** 10.3390/ijerph20075327

**Published:** 2023-03-30

**Authors:** Manuel-Jesús Jiménez-Roldán, Borja Sañudo, Luis Carrasco Páez

**Affiliations:** 1Department of Physical Education and Sport, University of Osuna, 41640 Osuna, Spain; 2Department of Physical Education and Sport, University of Sevilla, 41004 Sevilla, Spain

**Keywords:** HIIT, insulin growth factor-1, executive functions, sedentary lifestyle, young adult, female

## Abstract

Sedentary lifestyles have become a social problem, mainly among women. A sedentary lifestyle has been associated with poorer health in this population, negatively affecting physical and mental health. Physical exercise, in particular high-intensity interval training (HIIT), has been shown to be a neuroprotective tool. The present study provides a protocol design for a parallel-group Randomized Controlled Trial (RCT), whose aim will be to compare the effect of two physical interventions, HIIT and physical activity (increasing daily steps), on Insulin Growth Factor-1 (saliva IGF-1 concentrations), executive functions, quality of life, body composition, physical fitness, and physical activity in young sedentary women. At least 77 women will be recruited and randomly assigned to either a HIIT group (12-week exercise HIIT intervention, 3 sessions/week), the HIIT + PA group (12-week exercise HIIT intervention, 3 sessions/week, plus 10,000 steps/day), or a control group (usual care). The primary outcome measure will the chronic change in IGF-1 concentration levels measured in saliva. Secondary outcome measures will be: (i) executive functions; (ii) body composition; (iii) physical fitness; (iv) physical activity; and (v) quality of life. All outcomes will be assessed at the beginning of the study, after the intervention, and after three months of follow-up. After this intervention, we will be able to determine whether HIIT together with increased physical activity could be more effective than HIIT alone in IGF-1 stimulation. Furthermore, by comparing both intervention groups, we will be able to determine the differential effects on numerous health-related variables. Consequently, the conclusions of this study could help better understand the effects of a training program on IGF-1 concentration levels and executive functions. In addition, various strategies could be proposed through physical exercise to improve cognition in this age group, as well as to improve the health status of this sedentary population.

## 1. Introduction

A sedentary lifestyle, understood a lifestyle of activities that require minimal body movement [1]; is one of the main factors responsible for chronic diseases in adults. Focusing on the most common age group among university students, i.e., 18 to 25 years of age, there is a significant decrease in physical exercise at this educational stage [2]. This means that the physical exercise recommendations set out by the WHO [3] are not met. Furthermore, if we compare the sexes, women are found to be 8% more inactive than men [4]. 

For these reasons, this protocol study is proposed because it is interesting to investigate the possible health-related factors involved, such as quality of life, physical condition, and mental health. It is well known that sedentary behavior over time is correlated with a greater loss of muscle mass, an increase in fat mass [5], body fat percent and fat mass index [6], together with social and psychological changes. These conditions can have a premature negative impact on cognition, with deficits in executive functions and even a reduction in the concentration of key neurotrophic factors in the brain. In addition to this problem, COVID-19 has increased sedentary behavior in this population [7], with as many as 1.4 billon adults failing to meet the minimum recommendations proposed by the WHO [8]. This pandemic situation has also generated changes in the quality of life of this population, showing a worse perception of their general health and mental health status [9]. 

Young women are at high risk of being inactive and sedentary, which can alter their emotional state through increased anxiety, stress, reduced cognitive and executive function, and even increased all-cause mortality. In addition, the young adult period is a critical time for consolidating neural and synaptic pathways [10]. Therefore, reducing the time spent in sedentary positions may be fundamental for a good consolidation of cognitive abilities and, more specifically, of executive functions [11]. In this way, this strategy can lead to medium- to long-term prevention of cognitive impairment.

Physical exercise has proven to be a useful strategy to improve performance in executive functions in adults [12]. This positive impact may be due to its effect on plasticity and neuronal growth. In fact, physical exercise leads to increased cerebral blood flow, increased neurogenesis stimulation, optimized synaptic interconnections, and increased generation of neurotrophic factors such as IGF-1 [13]. IGF-1 has been shown to be a potent facilitator of brain plasticity in young adults. Thus, a higher concentration of IGF-1 may be conducive to higher brain performance and better cognition [14].

Despite these effects of physical exercise on executive functions and IGF-1 concentrations, it is known that, depending on the type of physical exercise performed, its effect may lead to different results. However, the intensity variable appears to be an important element in generating greater stimulation of IGF-1 [15]. Low–moderate intensity exercise has shown some chronic benefits in both IGF-1 concentration and executive functioning. In contrast, high-intensity exercise has shown greater performance for both variables. 

In recent years, due to the significant increase in the practice of HIIT in society [16], there has been a high level of interest in knowing its effect on different health variables, such as physical and psychological fitness [17], cardiometabolic parameters such as glycemic control, and even cognitive functions [15]. These studies show that HIIT is an efficient strategy to improve all these health-related variables. However, there is little evidence comparing the combination of a HIIT training program with an increase in light physical activity (daily steps) versus performing only the HIIT training program, or analyzing its effect on physical fitness, mental health, and quality of life in the population described.

Therefore, the hypothesis of the study is that a 12-week HIIT program implemented with an increase in daily physical activity (HIIT + PA) enhances to a greater extent executive functions, IGF-1 concentration, physical condition, body composition, and quality of life in sedentary young women than a similar program (HIIT) alone.

For resolve this issue, this plan includes a specific program focused on sedentary university women, with the objective of increasing their levels of physical activity. In this context, we designed an RCT to investigate the effects of a HIIT program versus the same program plus an increase in their daily physical activity on IGF-1 concentrations and executive functions. The purpose of this methodological article is to describe the study design, procedures, and methods that will be used in this research. 

## 2. Materials and Methods

### 2.1. Study Design

The present study is a parallel-group RCT (ClinicalTrials.gov accessed on 30 November 2022. ID: NCT05642169). The research study complies with the Declaration of Helsinki and has been reviewed and approved by the Andalusian Bioethics Committee (IC 1409-N-20). An organizational and participant flow diagram is shown in Figure 1, and the schedule of enrolment, interventions, and assessments is provided in Figure 2. 

### 2.2. Participants and Recruitment Process 

Women will be recruited through online dissemination through the institutional website of the University of Seville, emails, and social networks. A selection of all interested women will be made based on the inclusion and exclusion criterion detailed below (Table 1). Furthermore, they will have to provide their written informed consent before participating in the research, which will be carried out in agreement with the CONSORT statement [18].

### 2.3. Sample Size

The sample size will be calculated in an a priori analysis (G*Power V.3.1.9.6) on the basis of the change in IGF-1 levels with reference to the results of the Antonelli et al. study [22]. This study will require a sample size (for each group) of 37 (41, assuming a subject loss of 10%). This is for an alpha = 95%, a statistical power of 80% and a precision (d) of 5.23. According to the objectives and design of the study, and to form three groups of subjects, the subjects will be randomly distributed into two experimental groups (HIIT and HIIT + PA) and a control group (CG). All groups will be made up of healthy female university students who are all sedentary and inactive and willing to participate in the study.

To attain this number of female university students, recruitment will be carried out through online dissemination via the institutional website of the University of Seville, e-mails and social networks. Once all female students with interest in participation are registered, different information sessions will be scheduled prior to the start, during which the inclusion and exclusion criteria will be applied. In this way, a greater homogeneity of the sample will be obtained, and it will be possible to achieve a reduction of possible biases.

### 2.4. Randomization and Blinding

After baseline assessments, all participants will be randomly assigned to the HIIT intervention (HIIT), the HIIT intervention plus 10,000 steps per day (HIIT + PA), or the control group (CG). A simple computer-generated randomization sequence (www.randomizer.org) will be created prior to enrolling participants to assign them to the HIIT group, HIIT+PA, or the control group. While participants will be aware of their group allocation, outcome assessors and data analysts will be blinded to the allocation.

### 2.5. Intervention

Participants randomly assigned to the CG will not have any type of intervention. They will not be instructed to change their nutritional habits and will not to carry out any regular exercise program outside of the study; this will be not supervised. In relation to the two groups that will perform HIIT, the HIIT + PA group will have to complete 10,000 steps per day. In this way, they will focus on increasing their daily physical activity. In addition, this group will perform the same HIIT training program as the other experimental group. 

The training sessions will have a frequency of 3 times per week (Monday, Wednesday and Friday) and the duration will be between 39 to 54 min, depending on the training week. In each group there will be different trainers supervising the exercise program. The training sessions will be carried out online, with a trainer visualizing the physical actions to have good control of the intensities and to be able to provide feedback on the exercise technique. Thus, the exercise sessions will be designed, carefully supervised, guided, and instructed by qualified exercise professionals. Each session on the online platforms will be supervised, in which the trainers work according to the standardized protocol. Each session will comprise an initial warm-up and a final cool-down component. Before each workout, participants must fill out a questionnaire [23] to inform about their health and accumulated fatigue prior to the workout. At the end of the week, they will complete the PACES questionnaire [24] to indicate their enjoyment of the practice of physical exercise. 

In the warm-up part of the training session, which has a duration of 10 min, joint mobility and muscle activation exercises will be considered. This will then be followed by the main part of the training, which ranges between 24–36 min, where exercises will be given in a progressive and incremental manner. The session will end with a 5 min cool down with stretching and relaxation exercises. Additionally, the training program will be developed with the subjects’ own body, so that they can make the necessary adaptations to individualize the training loads. Therefore, a progression will be made according to the needs of this population. This intervention will meet the training standards of the American College of Sports Medicine [16]. For this, adaptations will be made both in the angulations and in the resistance involved in higher effort exercises. Throughout the main part of each workout, 30 s of work will be performed at the corresponding intensity followed by 30 s of rest. Halfway through the total repetitions, there will be 2 min of recovery.

The heart rate of the session will be measured with heart rate monitors (H10 Polar Electro OY, Finland) to control the exercise intensity. In addition, we will monitor perceived exertion (RPE) using the Borg scale (0–10). Both in the middle of the training, before rest, and at the end of the main part of the training, the RPE reported by each subject will be recorded. The training program will start with an intensity of RPE 8 in the first 3 weeks, reaching maximum intensity in the last 2 weeks of the training program. The programmed volume and intensity are shown in Figure 3. To maximize adherence, several strategies will be implemented, including individualized attention during the intervention sessions, telephone calls following non-attendance, and music in all sessions.

### 2.6. Primary and Secondary Outcomes

#### 2.6.1. Primary Outcomes

##### Insulin Growth Factor-1 (IGF-1) 

The main biochemical determinants established in this study will be measured in saliva. Thus, the magnitude of the IGF-1 response in saliva as a consequence of the execution of a 12-week HIIT training program will be analyzed. Furthermore, the acute response to maximum effort will be measured in the initial and final evaluations.

Saliva will be collected in a fasting state (between 8–12 h) and the passive drooling method will be used. At each assessment, a saliva sample will be obtained at rest and after the maximum effort. For this collection, participants will be asked to tilt their head forward and accumulate saliva. When they have a sufficient amount of saliva, they will be asked to drool in a 10 × 10 test tube. After collection, the samples will be centrifuged at 1650× *g* at 4 °C for 10 min. They will then be aliquoted and stored at −80 °C within one hour of collection. To measure the variability in each IGF-1 concentration assessment in saliva, the ELISA KIT protocol will be used.

Circulating IGF-1 levels (ng/mL) will be determined in saliva by an enzyme-linked immunosorbent assay (ELISA) and performed in duplicate. Samples will be added to the wells of the microtiter plate with the IGF-1 specific antibody. Horseradish Peroxidase (HRP)-conjugated Avidin will then be added to each well of the microplate and incubated. After adding the TMB substrate solution, only the wells containing IGF-1 will show a color change. This action will be measured spectrophotometrically at a wavelength of 450 nm ± 10 nm. The concentration of IGF-1 in the samples will be determined by comparing the concentration of the samples with the standard curve.

In this case, kits produced by Elabscience will be used, with the following characteristics (Table 2):

Before starting the measurement, the standards that will later be used in the design of the standard curve necessary for the calculation of the results must be reconstituted. For this purpose, 7 tubes with 0.5 mL of diluent buffer will be prepared and the diluted standard used to produce a double dilution series.

After thoroughly mixing each tube before the next transfer, a dilution series with 7 spots (100 ng/mL, 50 ng/mL, 25 ng/mL, 12.5 ng/mL, 6.25 ng/mL, 3.13 ng/mL, 1.56 ng/mL, and the last EP tube with diluent buffer is the blank at 0 ng/mL) will be prepared. In the development of the measurement, aliquots of 100 μL per well will be used, both in the case of the standards and the control sample as well as the saliva, previously thawed for 2 h at room temperature.

After a first incubation of 1 h at 37 °C, the solutions will be aspirated and each well washed with 350 μL of wash solution. The remaining liquid will be then completely removed from all wells by tapping the plate on absorbent paper. This washing will be performed 3 times. After the last wash, the plate will be inverted and dried on absorbent paper. Then 100 μL of detection solution B will be added, leaving it to incubate again for 30 min at 37 °C after covering it with the sealer. Subsequently, the washing process will be repeated a total of 5 times. After completion of the last wash, 90 μL of chromogenic substrate will be added to each well to cause staining of the solution. Again, it will be incubated for 15–25 min at 37 °C.

Finally, 50 μL of TMB stop solution will be added to stop the process and the plate will be read through the Biotek Synergy HT spectrophotometer (Biotek Winooski, VT, USA), adjusting the absorption to a wavelength of 450 nm. The Gen 5TM software (Biotek Winooski, VT, USA) will be used to analyze the data.

#### 2.6.2. Secondary Outcomes

Executive functions are composed of three skills, inhibitory control, working memory and cognitive flexibility. These, due to the function they perform, become essential skills for mental and physical health [25]. Cognitive performance, together with the executive functions, can be impaired due to the natural deterioration of human beings over the years. Therefore, it is important to know how the executive functions undergo various alterations with age. Thus, it is important to know the effect of physical exercise on executive functions as a preventive measure against early impairment.

Inhibitory control

For the evaluation of inhibitory control, the Stroop test (ST) will be used [26]. This is a neuropsychological test that measures the interference generated in the performance of a task. In this work, the Victoria version will be developed in a digitized form [23]. This is a short-duration test and consists of 3 parts (each of 24 stimuli) [27]. In the first part (D), the participant must press the appropriate buttons indicating the color (blue, red, yellow and green) of the circles presented on the monitor. In the second phase (W), the user must press the button corresponding to the color of the ink with which the word is written, which is different from the color name and does not generate any interference effect. Finally, a list of words indicating colors will appear, but these will be written in a different color than the one the word refers to. They must select the color of the ink, not the meaning of the word (C). This is the phase in which interference occurs. In addition, here the participant must inhibit the tendency to respond automatically.

In this test, the variables of time and the number of intrusion errors (C) due to inhibitory control failures will be analyzed. The ST has been shown to be a valid and reliable tool for application in the general population [28].

Cognitive flexibility

For this executive function, we will use the Wisconsin card test (WCST). This test mainly assesses cognitive flexibility [29]. It can be performed traditionally or, as in this case, digitally. This test consists of a total of 64 cards. The cards present different combinations according to 4 geometric shapes, 4 colors, and 4 quantities. The task begins with the psychologist giving an initial explanation of the different possible grouping options. The subject will observe on the screen the types of cards that she will have to keep in mind later for grouping.

At the beginning, the subject will have 4 reference cards, aligned in front of the person being evaluated. On the right, an image that simulates a deck will appear with the remaining cards placed face down. The subject, with the help of the mouse, should place each response card in front of one of the four reference cards where she thinks it should go, according to the initial logic applied by the computer program. After placing the card, it will appear to her whether it is “correct” or “incorrect”, but not where it should be placed.

The goal is to get as many correct answers as possible. For example, at the beginning, the cards should be placed according to color. When the subject gets 10 correct answers in a row, a category is reached and a category is changed, e.g., quantity, without warning. The following are the main variables derived from the digital version: number of correct responses and number of correct categories identified, perseverative responses (number of cards the subject sorts under a previous correct category, despite negative feedback from the experimenter), set maintenance failures (if the subject is sorting, for example, according to “color”, and forgets this category and changes the organization to another one, for example, “quantity”), perseverative errors (if the category is completed by placing the 10 cards correctly, and the subject continues placing cards according to the previous category), non-perseverative errors (those errors not corresponding to the two previous ones), total number of errors, single errors, and cards used to complete the 1st category.

This executive test has been investigated and shown to demonstrate high reliability and internal validity for measuring cognitive flexibility [29].

Working memory

The Digit Span Test (DST) is a subtest of the Wechsler Adult Intelligence Scale [30] that analyzes the ability to store and retain information, and its mission is to measure working memory. The test consists of two measures: (a) forward or direct; this consists of retaining direct digits, that is, the subject will be told a series of incremental digits presented orally, and must repeat them in the same order; and (b) backward or inverse; this consists of retaining digits inversely, in which the subject must repeat a sequence of incremental series of digits in inverse order to the one heard.

In both cases, the sequence will begin with the retention of the first series of length 3 (3 digits), which will be named orally by the examiner at a rate of one per second. If the participant succeeds in remembering the sequence correctly, the second series of 3 digits will be repeated. If this is answered correctly, the first series will proceed to the first series with length 4 and so on until the subject makes two consecutive errors in the same sequence of digits.

The result of each of the tests (forward and backward) corresponds to the number of digits reached of the last series that is attained without error, either in the first or second attempt. Therefore, the DST test offers the following results: (a) memory in direct order (DST-D), (b) memory in reverse order (DST-I-), and (c) memory total (DST-D+DST-I).

Body composition

Body composition will be measured by taking waist and hip perimeters and using electrical bioimpedance (BIA). This is a fast and non-invasive technique in which different body composition variables are estimated. This tool is a reliable system with scientific validity [31], although for its measurement, the situation in which the subject arrives must be standardized. For this, in this research, the subject had to follow the following guidelines:−Do not consume alcohol 24 h before the test.−Do not perform high-intensity physical exercise, and do not take caffeine or any other liquid or food 4 h before the test.−Do not perform the test immediately after getting up; it is advisable to perform the test two hours after getting out of bed so that body fluids are distributed normally.−Urinate 30 min before the test (on arrival at the laboratory).−Do not take diuretics 7 days before the test.−Do not wear metallic objects such as jewelry, watches, etc., at the time of the test.

The BIA used in this investigation (Tanita BC-418, Tokyo, Japan) is a segmental measurement. The subject is placed in a standing position on 4 electrodes, located on the feet and hands. An alternating current is introduced through two of these electrodes, and the remaining two collect this current, measuring, among others, the values of impedance, resistance and body reactance.

Among all the estimations made by the BIA, total body water can be determined. In addition, indirectly through hydration of the tissues, fat free mass is obtained and, by derivation, fat mass. On the other hand, muscle mass can be recorded. All these measurements will be recorded both in their total amount and in their equivalent percentage of the total. In addition to these variables, the mass, expressed in kilograms (kg) and bone mass (kg) will be determined.

Waist circumference (cm) will be evaluated at the midpoint between the ribs and the top of the iliac crest with the subject standing (Harpenden anthropometric tape, Holtain Ltd., Dyfed, UK). Complementarily, hip circumference (cm) will be measured at the level of the hips, specifically at the level of each greater trochanter.

Physical fitness

The physical fitness of the participants will be evaluated using the following tests.

Cardiorespiratory fitness:

The maximum oxygen consumption (VO_2_max) will be estimated from the maximum power output (MPO) achieved in an incremental cycloergometer test [32]. Therefore, only MPO will be taken into account for its estimation, since it is a measure of the capacity of the cardiorespiratory system, which can predict 80% of the variation in VO_2_max. Therefore, the calculation of VO_2_max will be applied according to the following equation [32]:VO_2_max (l·min^−1^) = (0.0117 × MPO) + 0.16

In order to perform the test correctly, the test conditions must be standardized. For this purpose, all tests will be performed after 8 h of fasting and with 20 min of prior rest. Additionally, all tests will be performed on the same cycloergometer (Ergoline model). This test has been shown to be an excellent alternative to apply in a young adult population, showing high reproducibility (r = 0.95) and good reliability (r = 0.88).

The development of the test begins with a first part of 7 min at a submaximal load of 69 W. At the end of this phase, the heart rate will be recorded with a pulsometer (H10 Polar Electro OY, Finalandia). The load will increase by 35 W every 2 min until the subject reaches exhaustion.

Muscle strength:

The strength of the leg extension muscle will be assessed on a calibrated isokinetic dynamometer (Biodex System 4, Biodex Medical System, Inc., Shirley, NY, USA). The Biodex System 4 will be used to assess each subject’s ability to generate maximal isometric power with the knee extensor musculature in a 60° flexion joint position from a seated position.

For measurement, the Biodex clinical data station will be used to analyze the Biodex torque and position signals and obtain the resulting measurements.

The positioning of the legs will be standardized for each subject so that it will always be assessed in the same position. The hip position will be kept flexed at 90° and shoulder and pelvic straps will be placed on each subject. In addition, the area of the thigh and tibia will be stabilized to provide additional thrust.

The performance protocol will consist of executing two maximal isometric efforts for 5 s. The subject, with arms crossed over the chest, will begin to perform the maximal force at the evaluator’s “push” command. Subsequently, the subject will rest for 30 s before performing the second attempt. Once both attempts have been completed, the best result of the peak torque variable (N-M) will be recorded for further analysis.

Upper extremity muscle strength:

Upper extremity muscle strength will be assessed by manual dynamometry testing with a digital dynamometer (TKK 5101 Grip-D; Takey, Tokyo, Japan) as described in other studies [33]. Subjects will be placed in a bipedal position during the test with arms in full extension and without touching any part of the body with the dynamometer, except the hand. The position of the body should be with the shoulder girdle parallel to the ground at all times, without allowing flexion or lateral movement of the trunk during the application of force.

However, the optimal grip of the dynamometer should be adjusted according to the size of the hand (maximum width between the first and fifth finger), measured in centimeters. The algorithm for women previously proposed [34] will be then applied: y = x/5 + 1.5, where X is the measurement taken previously. The result is the optimal grip amplitude to which the dynamometer will be adjusted before performing the test.

Once the dynamometer is adjusted, each subject will perform the test twice, alternating the execution hand and having 1 min of rest between both measurements. Participants will be instructed to squeeze the dynamometer grip as hard as possible and without bending any of the joints involved. The duration of the effort will be 5 s, without being able to perform discontinuous activations, i.e., they must maintain the maximum isometric force possible during the 5 s. Finally, the best result (kg) of the two tests performed will be recorded.

Physical activity and sedentary behavior (Accelerometry)

Physical activity will be objectively evaluated by accelerometry. The accelerometer is a device that allows objective quantification of the physical activity performed by a person. The study population will be asked to wear a triaxial accelerometer (GTX3, Penascola, FL, USA).

For both the initial setup prior to the evaluations and the data dump recorded during the evaluation week, Actilife software (Actigraph, V.6, Pensacola, FL, USA) will be used. The time spent in physical activity at light, moderate, and vigorous intensity and sedentary time will be calculated. The accelerometers will be programmed with a frequency of 60 Hz and an EPOCH of 15 s.

Participants will be instructed to wear the accelerometer throughout the day (24 h). For subsequent analysis, subjects must comply with at least 4 full days (8 h per day) of carry, one of them being a weekend, since the results obtained in 4 days of data collection can be extrapolated to 1 week [34].

The location of the accelerometer during recording days should be on the right side at hip height and in line with the anterior face of the ipsilateral lower extremity. To avoid damage to the device, it must be removed in any aquatic situation. On this occasion, a record sheet will be provided for its annotation.

International Physical Activity Questionnaire (IPAQ)

The IPAQ questionnaire is an instrument to determine approximately the level of physical activity practiced by the general population [35]. This questionnaire is one of the most widely used for the detection of sedentary lifestyles and is used to a greater extent in its short version, which has been adapted and translated into Spanish. Additionally, this questionnaire shows excellent reliability [18] and high validity for its application in the university population [36].

The IPAQ in its short, self-administered version consists of seven questions, covering four domains of physical activity: work-related, transportation (walking), housework/gardening, and leisure time activity. The questionnaire also includes questions on the time spent sitting as an indicator of sedentary behavior. In each of the four domains, the number of days per week and the daily time spent in moderate and vigorous activity are recorded.

According to these results, individuals can be classified into 4 categories. Category 1, or low level of physical activity, shows insufficient physical activity. Category 2, or moderate level of physical activity, shows 3 criteria to be classified as active:−3 or more days of vigorous physical activity for at least 20 min per day; or−5 or more days of moderate physical activity and/or walking a minimum of 30 min per day; or−5 or more days of any combination of moderate physical activity and/or walking and/or vigorous walking, reaching an energy expenditure of at least 600 Mets-min (basal metabolic units) per week.

Finally, category 3 includes very active people who achieve an energy expenditure of at least 1500 to 3000 Mets-min per week. For the calculation of Mets-min per week, we use the compendium of energy expenditure for the different physical activities of Ainsworth et al. Therefore, for the calculation of the IPAQ data, an average of Mets will be adopted for each type of activity as follows: walking, 3.3 Mets; moderate activities, 4.0 Mets; and intense activities, 8.0 Mets.

Quality of life (Short form SF-36)

To detect changes in health-related quality of life, the SF-36 questionnaire will be used. This questionnaire, in its Spanish version [37], focuses on assessing general health status through a generic scale. This provides a profile of quality of life for both a general population and specific subgroups.

This questionnaire consists of a total of 36 questions covering 8 health concepts. Additionally, there are two summary measures, which are physical health and mental health. Mental health is divided into the vitality scale (4 items), social functioning (2 items), emotional role (3 items), and mental health (5 items). On the other hand, physical health encompasses physical capacity (10 items), physical role (4 items), bodily pain (2 items), and general health (5 items). In addition, two global questions on the perception of the subject’s health have been added.

The scores for each scale range from 0 to 100, with higher scores indicating a better quality of life.

### 2.7. Statistical Analysis

Once data are obtained in the three research groups (HIIT, HIIT + PA, and CG) they will be compared at baseline, after 12 weeks of intervention, and three months after the end of the training program. Frequency tables will be obtained for the sociodemographic variables and means and standard deviations for the quantitative variables. The effects of the intervention on the primary and secondary outcomes will be evaluated with repeated measured analysis. The effect size (95% confidence interval), percentage of change, and statistical significance will be reported for each outcome with regards to the main group (between-subjects) time (within-subjects) and their interaction (group × time) effects.

Finally, various correlation analyses between the different variables will be carried out using Spearman’s Rho.

All data will be processed with the SPSS v.26 statistical program, establishing a confidence interval of 95% and estimating a significance level of *p* equal to or less than 0.05.

## 3. Discussion

This paper describes the protocol for an RCT that aims to determine the effect of a HIIT program on IGF-1 and other health-related outcomes in sedentary university women.

Although it is known that HIIT can improve IGF-1 concentrations and executive function performance [14], the effect of increasing daily physical activity on these variables is not well known.

To best of our knowledge, no HIIT interventions have been developed where a HIIT program is developed in conjunction with an increase in daily physical activity, with the aim of observing changes mainly in IGF-1 and executive functions in the population studied.

Previous studies with a similar methodology [38], but without focusing on increasing daily physical activity, have shown that 6 weeks of HIIT together with health education can reduce unhealthy behaviors and show changes in cognitive and health-related variables. In contrast, our study has a duration of 12 weeks, so this may allow us to determine if increasing daily physical activity can achieve a greater result in the concentration of IGF-1 and in the executive and health-related variables. Through our hypothesis, it is expected that an increase in daily steps together with the HIIT program will lead to a greater effect than doing HIIT alone or the combination of HIIT and health education.

After completing this research, we will have a detailed overview of whether this strategy is more effective than others previously studied, and also whether there has been a high adherence to the program, in view of the large increase in physical activity in the HIIT + PA group.

Thus, more research is needed to know the effect of physical exercise on sedentary girls, with or without the incidence of physical activity, to see the possible benefits to cognitive ability. This project could provide novel results that have been little studied to date, and that could also be generalizable to the sedentary female population in general. Furthermore, we could learn more about the relevance of physical activity as a complement to a physical exercise program on different variables related to health and cognition.

This intervention design has total reliability and scientific validity. Therefore, if the results of this RCT provide us with positive health benefits, it could become an excellent tool for integration in different settings and contexts with similar characteristics. In addition to the cognitive variables, this study relates the changes with other variables related to health, in order to understand the problem from different perspectives. Therefore, this project can provide relevant information for the general population and be a strategy to prevent subsequent diseases in the female population.

This said, the design of the study has various limitations. First, this project, in addition to IGF-1, could have more cognitive biomarkers (e.g., BDNF and β-amyloid levels) and brain imaging (e.g., fMRI) to learn more about the different underlying mechanisms. Second, since it involves an online training program, even if it is supervised, there may be difficulties in controlling the intensity and execution of the exercises. Even so, attempts will be made to monitor heart rate and RPE throughout the training sessions. Despite this limitation, an online training program offers a new perspective for the population. In addition, this type of low-cost intervention modality can benefit and extend to more people, so it can be an efficient strategy to implement for sedentary people.

## Figures and Tables

**Figure 1 ijerph-20-05327-f001:**
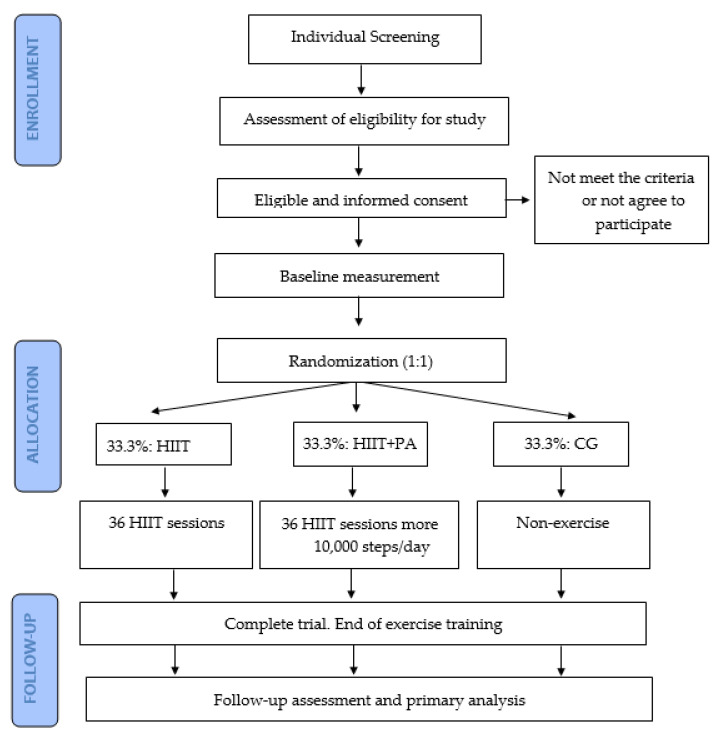
The organizational and participant flow diagram.

**Figure 2 ijerph-20-05327-f002:**
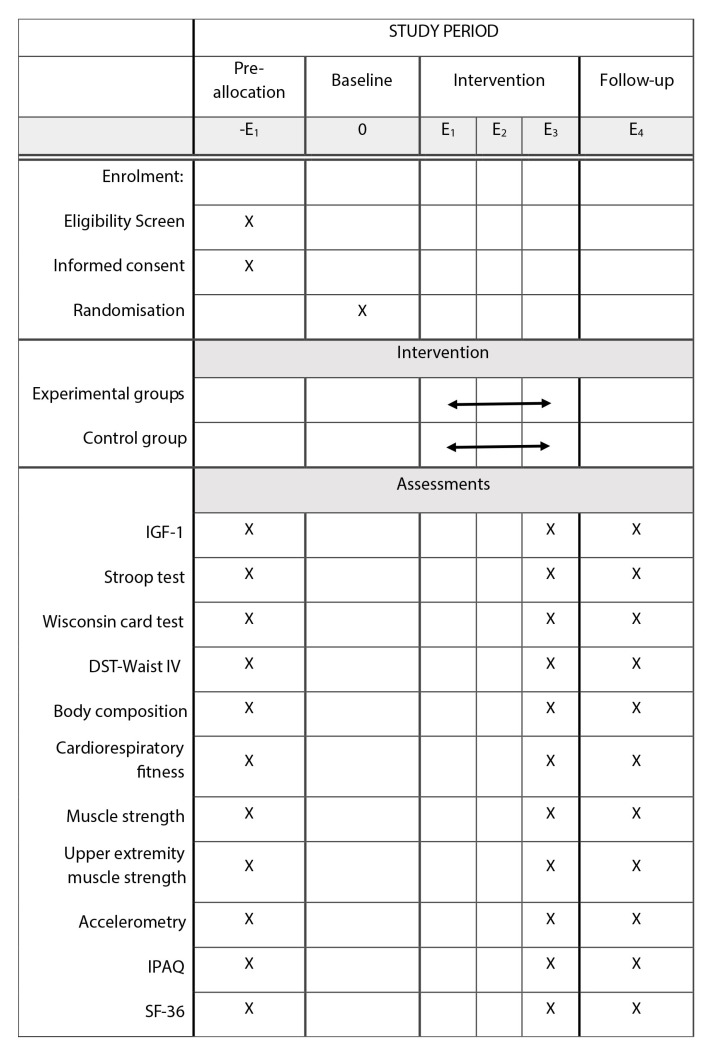
Schedule of enrolment, interventions and assessments in RCT.

**Figure 3 ijerph-20-05327-f003:**
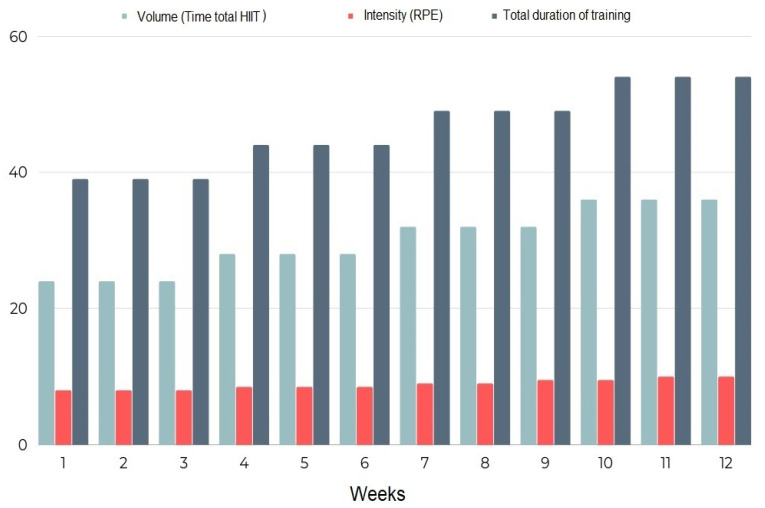
Intensity, volume, and total time of the session scheduled in the training program.

**Table 1 ijerph-20-05327-t001:** Study inclusion and exclusion criteria.

Inclusion Criteria
− Age: 18–30 years old − Indicate in the IPAQ questionnaire that they are in category 1 or low level of physical activity [19] and therefore do not conform to the minimum recommendations for physical activity and physical exercise proposed by the WHO. − Not suffering from somatic symptoms or serious psychiatric illnesses that prevent the practice of physical exercise. − Not attended psychological therapies in the last 12 months prior to the study. The use of this type of treatment may have an impact on improving cognitive abilities and executive functions of the study population. − Be able to communicate. − Informed consent: be able and willing to give informed consent.
**Exclusion criteria:**
− Having a general cognitive impairment or executive function disorder recognized and treated by specialists. − Physical activity: to select subjects who are sedentary, we excluded those who engage in physical activity in their leisure time in a continuous and planned manner. − Bilingualism: habitual and indistinct use of two or more languages. This may condition the results of various cognitive tests present in the study, since it is known that this intellectual capacity has a morphological and functional impact on the brain [20] − Musical instrument: playing a musical instrument regularly and continuously, as in the previous criterion, generates a positive effect on the subject’s cognition [21].

IPAQ: Physical Activity Questionnaire, WHO: World Health Organization.

**Table 2 ijerph-20-05327-t002:** Technical characteristics of IGF-1 ELISA kits produced by Elabscience.

Detection Range (ng/mL)	1.56–100 ng/mL
Sensitivity (ng/mL)	0.94 ng/mL
Specificity	This kit recognizes human IGF-1 in the sample.
Repeatability	The coefficient of variation is <10%.

CV: Coefficient of variation; mL: milliliters; ng: nanogram.

## Data Availability

The data presented in this study are available on request from the corresponding author.

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
