# Peer review of "Influence of High-Intensity Interval Training on IGF-1 Response, Brain Executive Function, Physical Fitness and Quality of Life in Sedentary Young University Women—Protocol for a Randomized Controlled Trial"

_ijerph, 2023, doi:10.3390/ijerph20075327_

Round 1
Reviewer 1 Report
Thank you for this interesting article. I would like to suggest a few points for modification:
1) The most serious problem that exists is that what you are trying to publish is the study protocol, but this is not made clear until the end of the theoretical framework. It is important that this point is made clear in the theoretical framework, abstract, etc. so that the reader knows from the outset what to expect.
2) What effects has COVID-19 had on the variables included in this research, especially on women? Please devote a paragraph in the introduction to this.
3) The theoretical framework omits some publications that could be important to set the context of the present study. Their reading and inclusion is recommended:
- https://doi.org/10.1007/s10964-021-01552-7
4) Why does it focus on adult population? This has not been clarified in the theoretical framework.
Author Response
We are very grateful to the reviewers for their careful reading of our manuscript and their constructive criticism. Our responses to the reviewer’s comments are written below and we have highlighted each change in the revised manuscript.

Reviewer 2 Report
Dear authors,
this is an interesting protocol paper wich may be useful for clinicians and other researchers in different contexts of study (chronic diseases, athletes...).
Could you please add some preliminary results to reinforce the importance of your protocol and make a proofread of your article by an native English speaker and some minor changes (e.g., line 20 "The primary outcome measure is" should written in the future (will instead of is) not in the present form.
Best.
Author Response

(The authors gave the same response as above.)
